# Hyponatremia and Cancer: From Bedside to Benchside

**DOI:** 10.3390/cancers15041197

**Published:** 2023-02-13

**Authors:** Benedetta Fibbi, Giada Marroncini, Laura Naldi, Cecilia Anceschi, Alice Errico, Dario Norello, Alessandro Peri

**Affiliations:** 1Endocrinology, Department of Geriatric Medicine, Careggi University Hospital, 50139 Florence, Italy; 2Pituitary Diseases and Sodium Alterations Unit, AOU Careggi, 50139 Florence, Italy; 3Department of Experimental and Clinical Biomedical Sciences “Mario Serio”, University of Florence, 50139 Florence, Italy

**Keywords:** hyponatremia, arginine vasopressin, cancer, syndrome of inappropriate antidiuresis, oxidative stress, tolvaptan

## Abstract

**Simple Summary:**

This review article is focused on hyponatremia in cancer patients. Hyponatremia is the most common electrolyte disorder in these patients and there is evidence that it negatively affects the course of the disease. Basic research has shown an increased proliferation rate and motility of cells cultured in low [Na^+^] conditions. Interestingly, the vasopressin receptor antagonist tolvaptan, originally approved for the treatment of hyponatremia secondary to the syndrome of inappropriate antidiuresis, and subsequently in polycystic kidney disease, counteracts cancer cell proliferation and invasivity in vitro. This result might encourage the design of clinical trials to determine whether tolvaptan may also have a role against cancer on clinical grounds.

**Abstract:**

Hyponatremia is the most common electrolyte disorder encountered in hospitalized patients. This applies also to cancer patients. Multiple causes can lead to hyponatremia, but most frequently this electrolyte disorder is due to the syndrome of inappropriate antidiuresis. In cancer patients, this syndrome is mostly secondary to ectopic secretion of arginine vasopressin by tumoral cells. In addition, several chemotherapeutic drugs induce the release of arginine vasopressin by the hypothalamus. There is evidence that hyponatremia is associated to a more negative outcome in several pathologies, including cancer. Many studies have demonstrated that in different cancer types, both progression-free survival and overall survival are negatively affected by hyponatremia, whereas the correction of serum [Na^+^] has a positive effect on patient outcome. In vitro studies have shown that cells grown in low [Na^+^] have a greater proliferation rate and motility, due to a dysregulation in intracellular signalling pathways. Noteworthy, vasopressin receptors antagonists, which were approved more than a decade ago for the treatment of euvolemic and hypervolemic hyponatremia, have shown unexpected antiproliferative effects. Because of this property, vaptans were also approved for the treatment of polycystic kidney disease. In vitro evidence indicated that this family of drugs effectively counteracts proliferation and invasivity of cancer cells, thus possibly opening a new scenario among the pharmacological strategies to treat cancer.

## 1. Introduction

Hyponatremia is the most common electrolyte disorder among hospitalized patients, with a prevalence of about 30% [1,2]. The aetiology of hyponatremia is multifaceted, but the most frequent cause is the syndrome of inappropriate antidiuresis (SIAD), which accounts for 40–50% of cases [3]. Severe, acute hyponatremia is associated to neurological symptoms, due to brain oedema, which may lead to brain stem herniation, respiratory arrest and death [2]. However, it is now well known that also chronic and mild hyponatremia may cause neurological as well as non-neurological alterations, such as gait disturbances, risk of fall, cognitive impairment and bone demineralization [4,5]. Moreover, it has been demonstrated that hyponatremia, if prolonged over time, can lead to an increased risk of death [6,7,8,9].

Hyponatraemia is the most frequent electrolyte alteration occurring in cancer patients. In this subgroup of patients hyponatremia is present in up to 40% of cases at the time of hospital admission [10,11]. Furthermore, about 50% of cancer patients with hyponatremia present, have one or more episode of mild to severe hyponatremia during the course of the disease, with a peak of 75% in patients with small cell (SCLC) or non-small cell lung cancer (NSCLC) [12]. Moreover, in this clinical setting, SIAD is the most frequent cause of hyponatremia [13]. However, other factors may determine hyponatremia in cancer patients, including associated comorbidities. Notably, hyponatremia has been associated to a significantly decreased progression-free and overall survival in these patients [10] and serum [Na^+^] has been proposed as a possible biomarker to identify high-risk cancer patients [14]. This review will provide an overview on hyponatremia in cancer patients, from aetiology to clinical manifestations and impact on patient outcome. A proper space will be dedicated to the only specific class of drugs approved for the treatment of hyponatremia secondary to SIAD, so far, namely vaptans, which are arginine vasopressin (AVP) and receptor (AVPR) antagonist. This review will cover clinical aspects as well as current evidence from basic research.

## 2. Hyponatremia in Cancer and AVP Receptor Antagonists

### 2.1. Aetiology of Hyponatremia in Cancer Patients

As already mentioned, hyponatremia in cancer patients can be determined by multiple causes.

Hyponatremia can be secondary to SIAD in patients with cancer of the lung (most frequently), nasopharynx, oropharynx, stomach, duodenum, colon, pancreas, prostate, uterus, ureter, bladder, breast or with lymphomas, leukaemias, sarcomas, brain tumours, mesotheliomas and thymomas [15,16,17]. In these cases, the pathogenesis is due to ectopic secretion of AVP [18]. Overall, the probability that hyponatremia in a cancer patient is due to tumoral AVP secretion is >30% [15].

Another cause of SIAD in these patients can be related to pharmacological treatments, which can induce hyponatremia by stimulating the release of AVP, or by potentiating the effect of AVP on the AVPR type 2 or by a mixed action [17,19]. Among these, chemotherapeutic drugs [e.g., cyclophosphamide, vincristine, vinblastine, cisplatin (although more frequently induces a salt loosing nephropathy), and melphalan], morphine and other narcotic painkillers, immunomodulators (immunoglobulins, interferon, interleukin-2, levamisole) [20], antiepileptic drugs (mostly carbamazepine and oxcarbazepine, but also eslicarbazepine, sodium valproate, lamotrigine, levetiracetam and gabapentin), antidepressants (e.g., selective serotonin reuptake inhibitors and tricyclics), phenothiazines that are used as antiemetic agents, and non-steroidal anti-inflammatory drugs deserve a mention. Recent therapeutic strategies, such as targeted therapies and treatment with immune-check points inhibitors, should also be included among the anticancer drugs responsible for hyponatremia, since they can cause corticotropin deficiency and consequently hypocortisolism [21,22].

Furthermore, in cancer patients, other conditions can induce or aggravate hyponatremia, such as nausea, vomiting, pain, physical and emotional stress, hydration in patients subjected to chemotherapy, diarrhoea, and heart or kidney failure [20].

### 2.2. Clinical Manifestations of Chronic Hyponatremia

Although it has long been thought that persistently, but mildly reduced serum [Na^+^], was not associated to clinical manifestations, and therefore did not require any correction [1], currently chronic hyponatremia is known to be not completely inconsequential on health, as previously mentioned. Indeed, even a slight decrease of serum [Na^+^] is able to induce adverse outcomes on several systems and body functions, if prolonged over time [23], thus explaining the raised morbidity and mortality observed in mildly affected patients [8,9,24,25,26]. Serum [Na^+^] ranging between 130 and 135 mEq/L is sufficient to determine a perturbation of internal homeostasis, which in turn can lead to short- and long-term complications and potentially life-threatening events. The correlation between mild chronic hyponatremia, hospitalization and increased mortality was confirmed in cross-sectional studies conducted in outpatient and inpatient settings [24,25], and in a meta-analysis including 81 studies and approximately 150,000 participants, which estimated a risk ratio for overall mortality equal to 2.60 (95% confidence interval [CI], 2.31–2.93) in hyponatremic compared to normonatremic subjects [7]. The presence of mild hyponatremia was also associated to a worse prognosis in patients concurrently affected by liver, kidney, heart and lung diseases [27,28,29,30], even if it was initially interpreted as a marker of disease severity rather than an independent risk factor able to potentially accelerate clinical deterioration [31]. Indeed, clinical settings such as cirrhosis and heart failure are characterized by an inadequate blood volume, which increases AVP secretion through non-osmotic triggers (in order to maintain adequate levels of blood pressure and circulating volume) [32,33], and in response to hypothalamic-pituitary-adrenal axis activation [34]. Anyway, the simultaneous measurement of copeptin—a neuropeptide co-released with AVP in equimolar concentrations and therefore considered a circulating surrogate marker of AVP activity—and serum [Na^+^] in a cohort of 6962 patients revealed a statistically significant correlation between all cause 30-day mortality and hyponatremia, even independently of copeptin levels [35].

As in acute hyponatremia, which is characterized by neurological symptoms due to brain oedema, the central nervous system is the main target of the chronic reduction of serum [Na^+^], too. Following a careful neurological evaluation, neurocognitive deficits are commonly recognized in apparently asymptomatic hyponatremic patients, as demonstrated in a cohort of 122 hyponatremic subjects admitted to an emergency department, that were characterized by frequent falls and unsteadiness higher than age-matched normonatremic controls [4]. Gait disturbances induced by low serum [Na^+^] are similar to those secondary to ethanol ingestion, are more severe in patients over 65 years than in younger subjects [36] and normalize after hyponatremia correction [4], thus suggesting a pathophysiological role of this electrolyte imbalance in neuronal functions. This correlation was confirmed more recently by the Osteoporotic Fractures in Men Study; the analysis of data from 5435 patients revealed a statistical association between mild hyponatremia, cognitive impairment and decline at the baseline, even stronger for the Trail Making Test Part B (which measures executive functions, such as cognitive flexibility, attention and inhibition control and working memory) than for the Mini-Mental Status Examination score [37].

In the last decade, the bone emerged as another target of chronic hyponatremia. Mild hyponatremic subjects encounter more frequently bone fractures than normonatremic ones [38,39,40], with a higher risk of hospitalization [25,33,41], and an increased length of stay in the hospital and mortality [42]. The higher frequency of falls is not the only determinant of fractures in these patients. Indeed, epidemiological data pointed out a higher prevalence of osteoporosis in hyponatremic patients compared to normonatremic ones, with an increased risk of bone demineralization directly proportional both to duration and severity of the electrolyte imbalance [43], even exceeding that related to smoking or corticosteroids treatment [40,44,45,46].

Most of the clinical manifestations secondary to chronic hyponatremia revert after an appropriate correction of the electrolyte imbalance, thus confirming the direct effect of low serum [Na^+^] on body homeostasis. In hyponatremic patients, the increase of serum [Na^+^] determines a significant reduction in mortality [47], and the treatment of severe hyponatremia with the AVPR type 2 antagonist tolvaptan improves gait disturbances [4], mental health [48], neurocognitive functions [49], Timed Up and Go test performance, nerve conduction speed and F-wave latencies [50]. The INSIGHT trial also highlighted the improvement of bone health after 22 days of tolvaptan administration [49].

Lung diseases are a frequent determinant of hyponatremia, which occurs in about 30% of patients affected by pneumonia [51]. By a retrospective analysis in COVID-19 patients during the first pandemic period, our group reported a 22.9% prevalence of hyponatremia at hospital admission, with worse respiratory performances (evaluated as P/F, i.e., the ratio between the partial pressure of oxygen in arterial blood PaO_2_ and the inspired oxygen fraction FiO_2_) and higher IL-6 levels in hyponatremic vs. normonatremic hospitalized patients [52]. The pro-inflammatory cytokine IL-6, which is able to induce AVP release by central and peripheral mechanisms [53,54,55,56], may represent the common denominator of acute respiratory insufficiency and hyponatremia secondary to SIAD. Hyponatremia has also been correlated with COVID-19 poor outcomes. A meta-analysis that included eight studies and 11,493 patients showed increased mortality, prolonged hospitalization, and severe COVID-19 (defined as severe pneumonia and/or needing intensive care unit support/invasive mechanical ventilation) in hyponatremic subjects compared to normonatremic ones, with an increase of post-test probability of worse prognosis from 16% (normal serum [Na^+^]) up to 33% (serum [Na^+^] < 135 mEq/L) [57]. Most recently, we demonstrated that early hyponatremia in COVID-19 patients is associated to the presence of laboratory and imaging parameters indicating a greater pulmonary and right-sided heart involvement at follow-up [52].

### 2.3. Hyponatremia in Cancer: Does It Affect Patient Outcome?

In the oncology setting, the prevalence of chronic hyponatremia is even higher than in patients hospitalized for all causes and it affects nearly half of cancer patients [10]. Although it is well established that hyponatremia can affect health care costs, primarily due to the need of hospitalizations and a doubled length of hospital stay compared to normonatremic cancer subjects, and quality of life [58,59], the correlation between low serum [Na^+^] and cancer progression-free and overall survival unexpectedly emerged only in the last decade. Hyponatremia was reported as an independent, negative prognostic factor in different solid and blood tumours, such as NSCLC [12], and SCLC [60,61], gastrointestinal cancers [62,63], lymphoma [64], hepatocellular carcinoma [65,66], renal cell carcinoma [67,68], prostatic and pancreatic carcinoma [69,70], biliary tract cancer [71], mesothelioma [71], multiregional upper tract urothelial carcinoma [72], and epithelial ovarian cancer [73].

Reduced serum [Na^+^] has an impact on survival at all cancer stages [61,74], with the highest rate of in-hospital mortality (OR = 2.05, 95%CI 1.67–2.53) in metastatic patients [75] and a hazard ratio for death almost three fold higher than in normonatremic oncologic subjects [10].

Another important aspect in the relationship between hyponatremia and life expectancy is the influence of low serum [Na^+^] on the response to cancer therapy. In a cohort of NSCLC patients treated with pemetrexed-platinum doublet chemotherapy, the presence of hyponatremia significantly reduced median progression-free survival of patients compared to the normonatremic group (6 months vs. 7 months; *p* < 0.05), even after adjusting for confounding factors. Hence, the authors concluded that pre-treatment serum [Na^+^] is a determinant prognostic marker in stage IIIb/IV patients, able to predict a differential response to treatment [76]. Similar results were obtained in metastatic renal cell carcinoma treated with targeted therapy (sunitibib, sorafenib) [77] or everolimus [78], in neuroendocrine neoplasms treated with peptide receptor radionuclide therapy [79], and in hepatocellular carcinoma treated with sorafenib [80]. Interestingly, in NSCLC patients treated with the EGFR inhibitor erlotinib a multivariate analysis revealed that hyponatremia was an independent predictive factor of non-response to therapy, in the same way as a poor performance status and the absence of EGFR mutations in the tumoral tissue [81].

Two large retrospective studies conducted on the Danish population not only confirmed that low serum [Na^+^] negatively correlated with all-cause mortality, but also with a higher risk to develop a neoplasm [9,82]. In other words, hyponatremic subjects seem to be more likely to get cancer, and if this electrolyte disorder is not properly treated, they have a higher risk of death.

These data may suggest that the correction of hyponatremia has a favourable role in the outcome of cancer patients. As a matter of fact, the normalization of reduced serum [Na^+^] was shown to increase overall and progression-free survival in 15 pre-treatment hyponatremic patients that underwent carboplatinum/etoposide regimen for SCLC compared to uncorrected ones [83]. Accordingly, in a group of 69 patients affected by NSCLC and hyponatremia, both overall and progression-free survival were significantly lower in those who had been maintained hyponatremic compared to those with normalized serum [Na^+^] (respectively 4.7 vs. 11.6 months and 3.3 vs. 6.7 months) [84]. These results were confirmed also in a small cohort of patients with different but extensive and terminal cancers (overall survival 13.6 months vs. 16 days, corrected vs. uncorrected hyponatremia), thus determining the option of receiving multiple lines of antineoplastic treatment [85]. Based on these data, hyponatremia has been proposed as a reliable biomarker of high-risk subjects with lung cancer [14].

### 2.4. Hyponatremia and Cancer: What Did Basic Research Tell Us?

In vitro and in vivo models of chronic hyponatremia demonstrated that reduced [Na^+^] has multiple effects on the homeostasis of different cell types. Barsony et al. demonstrated that sustained low extracellular [Na^+^] directly stimulated osteoclastogenesis and resorptive activity and that low [Na^+^], rather than low osmolality, triggered these effects. The induction of oxidative stress, associated to reduced ascorbid acid uptake by osteoclasts, was one of the main effects that were observed [86]. Furthermore, aged hyponatremic rats manifested decreased body fat, skeletal muscle sarcopenia and cardiomyopathy, more precisely increased heart weight and perivascular and interstitial fibrosis [87]. Recently, the development of liver steatofibrosis, which increased in parallel with the progressive reduction of serum [Na^+^] was described for the first time in a mouse model of chronic hyponatremia. Accordingly, the expression of proteins involved in lipid metabolism (SREBP-1, PPARα and PPARγ) and in myofibroblast formation (αSMA and CTGF) significantly increased. Interestingly, in agreement with the aforementioned experimental evidence of hyponatremia-triggered oxidative stress, heme oxygenase-1 overexpression (HMOX-1) was detected in sinusoidal cells and in particular in Kupffer and stellate cells [88]. It is worth mentioning that on clinical grounds non-alcoholic fatty liver disease represents the first step of a variety of alterations that lead steatohepatitis, fibrosis, and ultimately cirrhosis and hepatocarcinoma [89].

With regard to the relationship between hyponatremia and cancer, seminal studies were performed in an in vitro model of chronic hyponatremia, in which neuroblastoma cells (i.e., SK-N-AS and SHSY5Y cell lines) were cultured in the presence of low extracellular [Na^+^] [90]. An extensive microarray analysis detected the presence of 44 genes, whose expression was modified compared to cells grown in normal [Na^+^]. Among these, the highest increase in the expression level was again observed for the HMOX-1 gene. The HMOX-1 is an inducible stress protein with a metabolic function in heme turnover [91] and with potent anti-apoptotic and antioxidant activities in different cells, including neurons [92]. It is known that in carcinogenesis oxidative stress, through the involvement of reactive oxygen species, has a key role in promoting local invasiveness and metastatization, by inducing genomic instability and/or transcriptional errors [93], and by activating pro-survival and pro-metastatic pathways [94]. Recently, HMOX-1 induction was associated to the proliferation and invasivity of different human cancer cell lines cultured in low [Na^+^]: pancreatic adenocarcinoma (PANC-1), neuroblastoma (SK-N-AS, SH-SY5Y), colorectal adenocarcinoma (HCT-8), chronic myeloid leukaemia (K562) and SCLC (NCI-H69) cell lines [95,96]. Increased expression of HMOX-1 in cancer cell lines grown in low [Na^+^] suggests the role of oxidative stress as the possible molecular basis of hyponatremia-associated poorer outcomes in oncologic patients. We further demonstrated that all the above-mentioned human cancer cell lines changed their morphology in the presence of reduced extracellular [Na^+^] and became able to grow in the absence of a solid substrate and to degrade extracellular matrix, thus increasing their invasive potential (Figure 1). In addition, the activation of pathways involved in proliferation and invasion (RhoA, ROCK-1, ROCK-2) was observed. The activation of these pathways was paralleled by a deregulation of the cytoskeleton-associated proteins, resulting in the promotion of actin cytoskeletal remodelling and cell invasion. Interestingly, the RhoA/ROCK1-2 pathway has been suggested as a possible therapeutic target in cancer [97,98,99], due to its involvement in actin cytoskeleton and in the promotion of actin polymerization via cofilin phosphorylation.

### 2.5. AVPR Antagonists

The story of AVPR antagonists, collectively known as vaptans, started about 30 years ago. The discovery of a non-peptide AVPR type 2 antagonist (OPC-31260), which caused aquaresis in rats, was reported in 1992 [100], whereas one year later the same results were also reported in humans [101]. Subsequently, other non-peptide AVPR antagonists were developed and among these there were OPC-41061 and YM087, which were commonly known thereafter as tolvaptan and conivaptan, respectively [102,103,104,105].

Vaptans competitively block the binding of AVP to AVPR. Some of them selectively bind to type 2 receptors, some others also bind to type 1 receptors. The AVPR type 2 antagonists are expressed on the membrane of renal collecting duct cells. The binding of a vaptan to this receptor type inhibits the activation of AVP-dependent signalling pathways, ultimately blocking the synthesis and transport of aquaporin-2 water channels into the apical membrane of the cells [17,106]. Therefore, free water reabsorption is inhibited. As a matter of fact, AVPR type 2 antagonists induce aquaresis, which cause a reduction of urine osmolality and an increase of serum [Na^+^]. With regard to vaptans currently used in clinical practice, overall, two molecules are available in the United States and Europe, conivaptan (by injection) and tolvaptan (oral tablet) (Table 1). Conivaptan was approved by the US Food and Drug Administration (FDA) in 2005, and its use is indicated for the “treatment of euvolemic and hypervolemic hyponatremia in hospitalized patients”. Tolvaptan was approved by the FDA in 2009 and is indicated for the “treatment of clinically significant hypervolemic and euvolemic hyponatremia. In Europe, tolvaptan was approved by the European Medicines Agency (EMA), in 2009, but, in contrast with the US, the indication was limited to the “treatment of adult patients with hyponatremia secondary to SIADH”. Despite EMA approval, European guidelines for the management of hyponatremia do not recommend the use of vaptans [107], claiming that the risk of overly rapid correction and/or hepatotoxicity may occur. However, overly rapid correction is usually safely prevented by a correct use of the drug (see below) and hepatotoxicity has been sometimes observed in patients taking much higher doses, for a different indication (see at the end of this chapter), than those used for hyponatremia. Hence, on clinical grounds tolvaptan use is widely spread also in Europe [108].

The initiation of tolvaptan treatment should be considered only in hospitalized patients, in order to closely monitor serum [Na^+^] in the first few days and adjust the daily dose, if needed, before discharge. Because of their mechanism of action, vaptans are not indicated for the correction of serum [Na^+^] in hypovolemic hyponatremia. In clinical practice, hypertonic saline remains the recommended treatment in euvolemic/hypervolemic hyponatremic patients with severe symptoms (e.g., respiratory distress, seizures). However, vaptans can be considered in hyponatremic patients with mild (e.g., mild neurocognitive impairment) or moderate (e.g., confusion, disorientation, unsteady gait) symptoms. Urea is another possible therapeutic strategy and doses between 15 and 60 g/day are usually effective in correcting serum [Na^+^]. Urea ingestion increases the intake of osmotic solutes, thus promoting clearance of water. It is indicated among the possible options to treat hyponatremia by the specific European guidelines and U.S. recommendations, although the poor palatability and the development of azotemia at high doses have often limited its use [17,106]. It has to be added that, in contrast to tolvaptan, urea treatment can be initiated outside the hospital, it is as effective as tolvaptan in the long term [109], it is less expensive [110] and it is also effective in the rare cases of nephrogenic SIAD, caused by gain-of-function mutations of the AVPR type 2 gene [111]. Effective correction of serum [Na^+^] with urea in cancer patients has been reported [110,112,113,114]. In mild hyponatremia an alternative treatment remains in principle fluid restriction. However, it is known that such a strategy is usually scarcely effective or is not tolerated [115]. It must be added that, in some instances, fluid restriction is not expected to give any result, in terms of serum [Na^+^] increase. This happens when the kidneys are not able to excrete solute free water, according to the urine/serum electrolyte ratio [116]. It is finally worth mentioning that in cancer patients receiving chemotherapy fluid restriction may be impossible to pursue, because of the need of appropriate hydration.

Vaptan treatment can be recommended in surgical and neoplastic patients with hyponatremia, with the aim to obtain a rapid correction and to proceed with surgery and/or chemotherapy without any delay. With regard to this point, it has been shown that pre-treatment hyponatremia is associated to a reduced survival in oncology patients [117]. Particular attention should be reserved to patients who must start chemotherapy, because several drugs can induce or worsen hyponatremia, as mentioned previously in this review. For completeness, considering that this manuscript is focused on hyponatremia in cancer, it is worth mentioning that another vaptan, mozavaptan, was approved in 2006 in Japan with the specific indication “for the treatment of patients with paraneoplastic SIAD” [118].

Interestingly, vaptans (OPC-31260 and tolvaptan) have been found to be effective in reducing cystogenesis in polycystic kidney disease (PKD) models, by reducing cAMP levels in kidney epithelial cells [119,120,121]. Multiple clinical studies, the Tolvaptan Efficacy and Safety in Management of ADPKD and Outcomes (TEMPO) program, have investigated the effects of tolvaptan in autosomal dominant PKD (ADPKD) and the results of a phase 3, multicentre, double-blind, placebo-controlled, 3-year trial were published in 2012 [122]. It was shown that tolvaptan (average dose, 95 mg/d) had a significant effect in counteracting the increase of kidney volume and the progression to end stage kidney failure, compared to placebo. Subsequently, tolvaptan effectiveness in the treatment of ADPKD was further assessed and currently the effect of this molecule in the inhibition of the cAMP/PKA pathway, thus contrasting the progression of the disease, is widely used in nephrology. Nevertheless, interventions targeting other signalling pathways downstream from cAMP/PKA might provide additional beneficial effects in the future [123].

### 2.6. Vaptans and Cancer: The Lab Perspective

As mentioned before, vaptans can bind the AVPR type 2 and block the cAMP-dependent intracellular signalling. In PKD rats, a model orthologous to human ADPKD, the suppression of plasma AVP by simply increasing water intake reduced renal cAMP and decreased the level of ERK activation, cell proliferation, and disease progression [124]. Therefore, PKD rats were selectively created with a non-sense mutation of the AVP gene and authors observed a reduced renal cAMP accumulation, ERK activity, cell proliferation, fibrosis and kidney were essentially free of cysts [121]. Consequently, administration of desmopressin, an AVP analogue, by osmotic minipumps, restored cystic disease in the AVP-deficient PKD rats, providing unequivocal evidence for the roles of AVP and cAMP on PKD progression. Recently, it has been demonstrated that the use of tolvaptan doses, likely to be attained in the plasma of ADPKD patients, inhibited AVP-induced proliferation through the B-Raf/MEK/ERK pathway and decreased both AVP-stimulated chlorine secretion and in vitro cyst growth of ADPKD cells [125]. In this way, the antiproliferative role of tolvaptan was hypothesized as a possible tool to be used in an oncological setting. For this reason, several in vitro and in vivo studies have been developed to test whether vaptans could be proposed in clinical practice both for the correction of serum [Na^+^] and possibly for counteracting cancer progression. According to The Cancer Genome Atlas (TCGA) and The Human Protein Atlas (THPA) databases, the expression of AVPR type 1 and type 2 is upregulated in several tumour types (breast, bladder, colon, lung, liver, ovary, pancreas, prostate, skin, thyroid, thymus, head and neck, sarcoma, and diffuse large B-cell lymphoma), with the greatest overexpression in kidney cancer. The ADPKD studies underlined an increased risk of liver injury in treated patients [122]. Therefore, the hepatoxicity of tolvaptan on human hepatocarcinoma cell line was analysed [126]. The authors found that tolvaptan inhibited cell growth, delayed cell cycle progression, caused DNA damage, and triggered apoptosis in HepG2 cells. We also obtained similar results in different cell lines from SCLC (NCI-H69), neuroblastoma (SK-N-AS) and colon adenocarcinoma (HCT-8), that express the AVPR type 2 [96,127]. We found that a 48-h exposure to tolvaptan significantly down regulated the expression of cAMP, PKA and AKT, decreased cell proliferation and invasion, with an IC_50_ in the micromolar range, probably through the secretion of type IV collagenases (MMP2 and MMP9), and inhibited the RhoA/ROCK1-2 pathway (Figure 2 and Appendix A)). In 2020, Sinha et al. carried out a study on renal carcinoma cells (clear cell Renal Carcinoma Cells, ccRCC), demonstrating in vitro and in vivo that AVPR type 2 is expressed in these cells and plays a pathological role in inducing tumour growth. It was demonstrated that tolvaptan and another AVPR type 2 antagonist, OPC31260, decreased ccRCC tumour growth by reducing cell proliferation and angiogenesis, whereas apoptosis was increased [128]. The ccRCC cancer is the most frequent kidney cancer subtype and it can be highly invasive. Commonly, patients affected by ccRCC have an unfavourable prognosis because of high recurrence rate after surgical resection, and neither chemotherapy nor radiation therapy are effective for patients with metastases [129,130,131]. Mozavaptan and tolvaptan were proposed as therapeutic strategies for the treatment of ccRCC and in vitro these molecules effectively reduced cell viability and clonogenicity in two ccRCC human cancer cell lines (i.e., 786-O and Caki-1 cells) by inducing cell cycle arrest. In addition, in a xenograft model of ccRCC, selective AVPR type 2 antagonists suppressed tumour growth by reducing cAMP levels and ERK activity [128]. The AVPR type 1 subtype has been also addressed as a target for antiproliferative strategies. Relcovaptan (SR-49059) is a non-peptide AVPR type 1a antagonist [132,133,134], which has shown positive results for the treatment of Raynaud’s disease and dysmenorrhea [135,136], even if it is not approved for clinical use. High levels of expression of AVPR type 1a were found in castration-resistant prostate cancer (CRPC), an incurable form of cancer that recurs after androgen deprivation therapy, and relcovaptan has been proposed as a possible cytolytic drug in this condition [137]. Preliminary in vitro studies showed that relcovaptan decreased cell proliferation by downregulating cyclin A, a key protein involved in the G2-M cell cycle transition. In contrast, AVP treatment in CRPC cells triggered cancer progression by the activation of ERK and CREB. Three preclinical mouse models supported the in vitro evidence that relcovaptan reduced CRPC growth, although xenografts did not appear to regress in any of the three models. A distinctive feature of prostate cancer is the occurrence of distant bone metastases [138] and relcovaptan protected against the formation of bone lesions. Following the results obtained in their previous study [137], in the last year, the authors demonstrated that there is a strong correlation between co-expression of AVPR type 1a and 2 and prostate cancer progression. Specifically, in CRPC xenografts the inhibitory effects of tolvaptan and recolvaptan on AVRP type 2 and 1a, respectively, significantly reduced tumour growth and the synergistic use of the two vaptans promoted cell apoptosis [139].

These studies revealed a new role of AVPR in the growth and invasivity of several tumour types and suggest AVPR antagonists as potential cytostatic and antiproliferative pharmacological strategies to slow down cancer progression.

## 3. Conclusions

There is strong evidence that hyponatremia has an important role in affecting patient outcome in cancer and that it cannot be neglected. In fact, hyponatremia should be promptly recognized and corrected, because the normalization of serum [Na^+^] has an independent beneficial effect on the prognosis of cancer patients. In this scenario, the encouraging effects of AVPR antagonists in counteracting cell proliferation and invasivity in experimental models could be considered, in the current absence of epidemiological data from cancer patients, in order to design specific trials on clinical grounds.

## Figures and Tables

**Figure 1 cancers-15-01197-f001:**
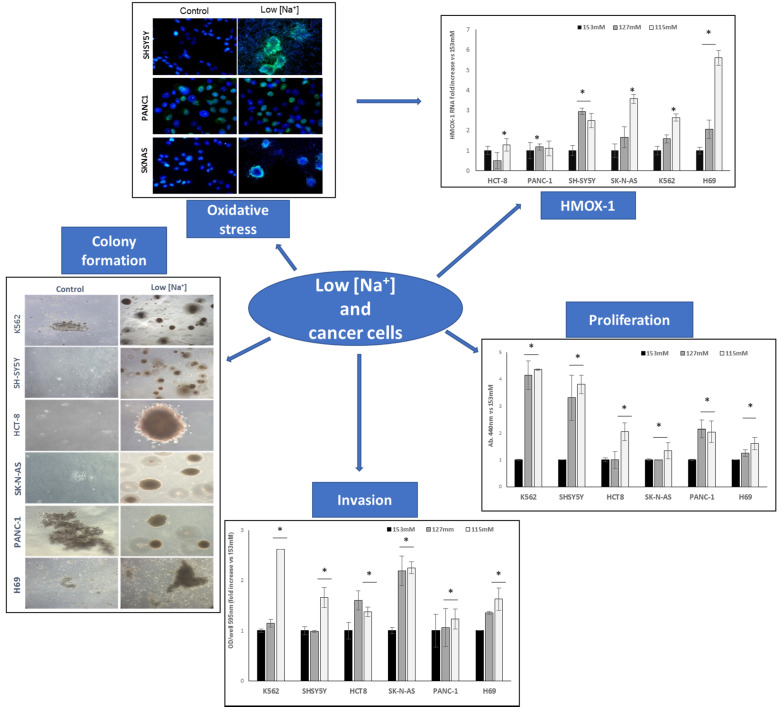
Different features of cancer cell lines cultured in normal extracellular [Na^+^] vs. low [Na^+^]. *: *p* ≤ 0.05.

**Figure 2 cancers-15-01197-f002:**
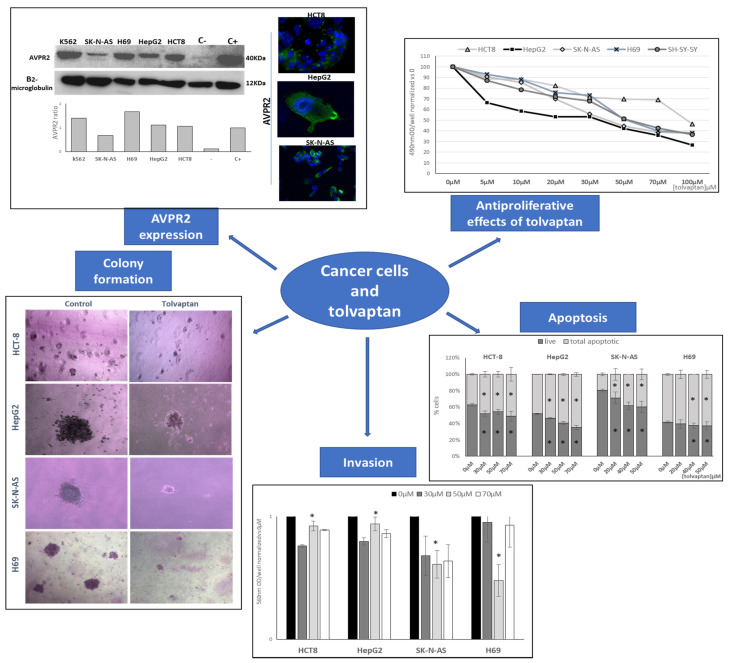
Effects of tolvaptan in cancer cell lines. *: *p* ≤ 0.05.

**Table 1 cancers-15-01197-t001:** Tolvaptan and conivaptan.

	Tolvaptan	Conivaptan
RECEPTOR	AVPR type 2	AVPR type 1a and 2
SELECTIVITY (K_i_V_1a_:K_i_V_2_)	29:1	10:1
ROUTE OF ADMINISTRATION	Oral	Intravenous
URINE VOLUME	Increased	Increased
URINE OSMOLALITY	Decreased	Decreased

K_i_ = dissociation constant of the inhibitor.

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
