# Peer review of "Hyponatremia and Cancer: From Bedside to Benchside"

_cancers, 2023, doi:10.3390/cancers15041197_

Round 1

Reviewer 1 Report

Very nice review on the subject

Some points deserve corrections:

- line 87Cisplatin induce more frequently a salt loosing nephropathy (this must be added.(treatment need frequently an increase in NaCL combined or not with fluorohydrocortisone.

_Antiepileptic drugs are also frequently associated with hyponatremia (particularly carbamazepine,etc)(to added)

_Concerning Urea treatment some comments are necessary

   - First inital treatment with urea can be done outside the hospital (no hospitalisation are necessary)

-In the long term urea has the same effectiveness as Tolvaptan (Soupart A et al CJASN )

-Urea is much cheaper than Tolvaptan (Eva Perello Camacho et al 2022)

-Urea is also effective in NSIAD (Decaux G JASN)whatever the mutation

-A rapid normalization of SNa is also easy to obtain with urea 

-Many studies have reported the treatment of SIADH associated with cancer

by urea (Nervo et al Clin Endocrinology 2019;Woudstra J et al Neth J Med 2020,Eva Perello Camacho et al 2022,C Sousa et al revistanefrologia 2022

Normalisation of SNa by urea or a vaptan will decrease cell proliferation but vaptans have in vitro an additionnal benefits ....Clinical studies are urgently needed

Author Response

REPLIES TO REV 1

Very nice review on the subject

We thank the reviewer for appreciating our manuscript

Some points deserve corrections:

- line 87 Cisplatin induce more frequently a salt loosing nephropathy (this must be added.(treatment need frequently an increase in NaCL combined or not with fluorohydrocortisone.

This detail has been added in the revised manuscript. (Chapter 2.1)

_Antiepileptic drugs are also frequently associated with hyponatremia (particularly carbamazepine, etc) (to added)

The reviewer is certainly right. We had missed to add antiepileptic drugs in the list. We have now added the name of antiepileptic drugs that can induce hyponatremia. (Chapter 2.1)

_Concerning Urea treatment some comments are necessary

- First inital treatment with urea can be done outside the hospital (no hospitalisation are necessary)

-In the long term urea has the same effectiveness as Tolvaptan (Soupart A et al CJASN )

-Urea is much cheaper than Tolvaptan (Eva Perello Camacho et al 2022)

-Urea is also effective in NSIAD (Decaux G JASN) whatever the mutation

-A rapid normalization of SNa is also easy to obtain with urea 

-Many studies have reported the treatment of SIADH associated with cancer by urea (Nervo et al Clin Endocrinology 2019;Woudstra J et al Neth J Med 2020,Eva Perello Camacho et al 2022,C Sousa et al revista nefrologia 2022

Normalisation of SNa by urea or a vaptan will decrease cell proliferation but vaptans have in vitro an additional benefits ....Clinical studies are urgently needed

According to the reviewer suggestions, we added sentences regarding the use of urea to correct hyponatremia. (Chapter 2.5)

Reviewer 2 Report

The authors conduct a review article and focused on hyponatremia in cancer patients.

Comments:

In vitro and in vivo models of chronic hyponatremia demonstrated that reduced [Na+] has multiple effects on the homeostasis of different cell types.

2.5. AVPR ANTAGONISTS(Table 1: tolvaptan and conivaptan)

Figure 1 Different features of cancer cell lines cultured in normal extracellular [Na+] vs low [Na+]

Figure 2 Effects of tolvaptan in cancer cell lines

Are there epidemiological data to provide such information (tolvaptan in cancer for human data)?

A summary table of epidemiological data(human) maybe could be provided in this review article.

Is apoptosis information missing (% cell) in Figure 2?

Author Response

REPLIES TO REV 2

The authors conduct a review article and focused on hyponatremia in cancer patients.

We thank the reviewer for appreciating our manuscript

Comments:

In vitro and in vivo models of chronic hyponatremia demonstrated that reduced [Na+] has multiple effects on the homeostasis of different cell types.

2.5. AVPR ANTAGONISTS (Table 1: tolvaptan and conivaptan)

We corrected the title of the chapter and the legend in Table 1

Figure 1 Different features of cancer cell lines cultured in normal extracellular [Na+] vs low [Na+]

We corrected the title of the figure

Figure 2 Effects of tolvaptan in cancer cell lines

We corrected the title of the figure

Are there epidemiological data to provide such information (tolvaptan in cancer for human data)?

A summary table of epidemiological data(human) maybe could be provided in this review article.

There are not epidemiological data regarding tolvaptan as an anti-proliferative drug in humans, yet, as added in the Conclusions.

Is apoptosis information missing (% cell) in Figure 2?

The fig has been redrawn

Reviewer 3 Report

The manuscript entitled: Hyponatremia and cancer: from bedside to benchside, is an overview about causes, clinical manifestations, consequences and therapeutic solutions of hyponatremia in patients with cancers. An important chapter is that about basic research studies discussing antiproliferative role of vaptans. This manuscript is important for clinical practice and for future research studies. It is well written.

Minor revision

I suggest reconsidering the following phrase from conclusion: ``It is the wish of the authors that at the end of this review the readers acknowledge that…``.  It is not so scientifically this formulation.

There are some minor editing errors like: ``In fact, hyponatremia should be promptly recognized ad corrected,…``

Author Response

REPLIES TO REV 3

The manuscript entitled: Hyponatremia and cancer: from bedside to benchside, is an overview about causes, clinical manifestations, consequences and therapeutic solutions of hyponatremia in patients with cancers. An important chapter is that about basic research studies discussing antiproliferative role of vaptans. This manuscript is important for clinical practice and for future research studies. It is well written.

We thank the reviewer for acknowledging the quality of our manuscript.

Minor revision

I suggest reconsidering the following phrase from conclusion: ``It is the wish of the authors that at the end of this review the readers acknowledge that…``.  It is not so scientifically this formulation.

This sentence has been erased in the Conclusions.

There are some minor editing errors like: ``In fact, hyponatremia should be promptly recognized ad corrected,…``

Editing errors have been corrected

Round 2

Reviewer 2 Report

No further comment